# Incidences of Fatalities on Austrian Ski Slopes: A 10-Year Analysis

**DOI:** 10.3390/ijerph17082916

**Published:** 2020-04-23

**Authors:** Markus Posch, Alois Schranz, Manfred Lener, Martin Burtscher, Gerhard Ruedl

**Affiliations:** 1Department of Sport Science, University of Innsbruck, A-6020 Innsbruck, Austria; 2Medalp Sportclinic, A-6460 Imst, Austria

**Keywords:** ski fatalities, Austrian ski slopes, traumatic, nontraumatic

## Abstract

The study evaluated incidences and potential differences of traumatic and nontraumatic fatalities among recreational skiers and snowboarders on Austrian ski slopes within a 10-year analysis. Within this retrospective study, data were collected by the Federal Ministry of the Interior. Data comprised all traumatic and nontraumatic deaths on Austrian ski slopes which occurred between the 2008/09 and 2017/18 winter seasons. Age, sex, nationality, gear used, altitude, slope difficulty, accident cause, primary cause of death and helmet use were collected at the death scene. Incidence of fatalities was calculated based on number of skier days. In total, 369 fatalities, with an average of 36.9 ± 7.9 fatalities per year, were registered. The yearly incidence of traumatic and nontraumatic deaths decreased by 25.8% and 40.1% during the 10-year time period, leading to an evaluated mean incidence of 0.70 deaths per million skier days, with an incidence of 0.36 traumatic deaths and 0.34 nontraumatic deaths per million skier days. Incidences of both traumatic and nontraumatic deaths decreased during the 10-year analysis, representing death as a rare event on Austrian ski slopes. However, adequate prevention measures to reduce potential risk factors to further reduce the mortality risk on ski slopes are needed.

## 1. Introduction

As more than 8 million skiers and snowboarders visit the Austrian Alps annually [1], recreational winter sport activities are associated with a certain risk of injury or death.

The overall incidence of ski injuries has decreased from 5–8 injuries per 1000 skier days [2] to 2–3 injuries per 1000 skier days in the 1990s [3,4] and the present evaluated injury rate in Austria is less than one injury per 1000 skier days [5]. In addition to non-fatal skiing-related injuries, ski fatalities also occur on ski slopes and a distinction is made between traumatic (e.g., collision with an object/person) and nontraumatic deaths (e.g., cardiac death) [6].

Regarding traumatic and nontraumatic fatal injuries on ski slopes, a rising trend of deaths rates could be shown between the period 1980 and 2001 by Xiang and Stallones [7] and death rates ranged from 0.53 to 1.88 deaths per million skier days. Later, after the turn of the millennium, the number of traumatic deaths among skiers has remained relatively stable, with incidences ranging from 0.37 [6] to 0.70 [8] and 0.75 deaths per million skier days [9]. Due to the lack of published literature/studies, the development of nontraumatic death incidences in the last two decades cannot be described properly.

In Austria, a study [6], evaluating incidences of fatalities between the years 2005 and 2010 revealed that the overall incidence of fatalities is 0.79 deaths per million skier days with an incidence of 0.42 nontraumatic deaths and 0.37 traumatic deaths per million skier days. Nowadays, the present evaluated injury rate in recreational alpine skiing in Austria is 0.6 injuries per 1000 skier days [5]. Therefore, ski fatalities on ski slopes are generally rare events compared to non-fatal skiing-related injuries [9]. When comparing fatal events on ski slopes with other mountain sport activities, the death risk during mountain hiking in the Alps is about five times, and during ice and rock climbing in the Alps or trekking in Nepal, about 10 times, higher than during alpine skiing [10].

Although a study by Ruedl et al. [6] reported of an average of 41 fatalities per year, the literature about ski fatalities and causes of fatalities among recreational skiers and snowboarders on Austrian ski slopes is scarce [6], especially long-term data on traumatic and nontraumatic fatalities for Austria.

Burtscher and Ponchia [11] reported that the risk of severe cardiovascular adverse events, accounting for most nontraumatic deaths [12,13], increases sharply in men over the age of 35 suffering from coronary artery disease, particularly those with prior myocardial infarction, and/or risk factors like arterial hypertension, hypercholesterolemia, or diabetes. To the best of our knowledge, no study investigated the potential risk factors regarding both traumatic and nontraumatic deaths on ski slopes. Furthermore, the majority of the available literature does not include data on nontraumatic deaths on ski slopes [8,9,14].

Thus, the goal of this study was to evaluate current incidences and potential differences of traumatic and nontraumatic fatalities among recreational skiers and snowboarders on Austrian ski slopes within a 10-year period from the winter seasons 2008/09 to 2017/18.

## 2. Materials and Methods 

In this retrospective study data were collected by members of the Federal Ministry of the Interior (FMI), who are qualified alpinists and have paramedical training. Data comprised all fatalities, including traumatic and nontraumatic deaths on Austrian ski slopes which occurred between the 2008/09 and 2017/18 winter seasons. Age, sex (male, female), nationality (Austrian, German, others), gear used (ski, snowboard, others), altitude, slope difficulty (easy, moderate, hard, unknown), accident cause (fall, collision, avalanche, sudden cardiac death, others), primary cause of death and helmet use (yes, no) were collected at the death scene and stored by the FMI and the Austrian Kuratorium für Alpine Sicherheit (KFAS). Causes of death were always determined by emergency physicians on the accident scene using information from companions and eyewitnesses. Incidence of fatalities was calculated based on the number of skier days during this 10-year period, which was offered by the Federation of Austrian ski lift companies.

This study was performed in conformity with the ethical standards of the 2008 Declaration of Helsinki. Furthermore, this study was approved (approval ID-62/2019) by the institutional review board (IRB) of the Department of Sport Science as well as the Board for Ethical Issues (BfEI) of the University of Innsbruck.

### Statistics

Data are presented as means and standard deviations, as well as absolute and relative frequencies. Factors with more than two categories (slope difficulty) were binary coded by making dummy variables for every single category to achieve univariate odds ratio (OR). Adjusted ORs and their 95% CIs for nontraumatic death are reported. To evaluate the changes of incidences of traumatic and nontraumatic deaths (winter seasons 2008/09–2017/18), the percentage of changes were calculated. According to the tests of normal distribution (Kolmogorov Smirnov) differences between traumatic and nontraumatic deaths in age and altitude were evaluated either by independent *t*-tests or Mann–Whitney-U tests. Differences between traumatic and nontraumatic deaths in frequencies (sex, gear used, slope difficulty) were computed by chi-square tests. SPSS 24.0 (IBM Corporation, Armonk, NY, USA) was used for the statistical analysis. All *p*-values were two-tailed and values of *p* < 0.05 were considered to indicate statistical significance. 

## 3. Results

In total, 369 fatalities during the time period between the winter seasons 2008/09 and 2017/18, with an average of 36.9 ± 7.9 fatalities per year, were registered (Figure 1). 

Most of fatalities occurred in males (87.3%) and in skiers (95.5%). Mean age was 51.0 ± 18.3 years and average altitude at time of death was 1552.0 ± 674.1 m above sea level. The majority of victims were citizens of Germany (40.7%) followed by Austrians (31.7%) and people from other countries (27.6%).

The evaluated mean incidence during this 10-year period (2008/09–2017/18) was 0.70 deaths per million skier days, with an incidence of 0.36 traumatic deaths and 0.34 nontraumatic deaths per million skier days (Figure 2). 

From the winter season 2008/09 to 2017/18, the yearly incidence of traumatic deaths decreased (−25.8%) from 0.37 (2008/09) to 0.27 deaths per million skier days, while the incidence of nontraumatic deaths decreased (−40.1%) from 0.46 to 0.27 deaths per million skier days. The highest incidences were reported in the winter season 2010/11 for traumatic (0.63 deaths per million skier days) and in 2009/10 for nontraumatic deaths (0.51 deaths per million skier days). The lowest incidences of 0.24 traumatic deaths were reported in 2011/12, whereas incidences of 0.22 nontraumatic deaths per million skier days were lowest in 2012/13. The course of traumatic death incidence was similar to that of the overall death incidence, whereas a different trend was shown among nontraumatic death incidence, especially between the period from 2008/09 to 2012/13 compared to overall death incidence (Figure 2). 

In sum, 190 (51.5%) were traumatic deaths and 179 (48.5%) were nontraumatic deaths. The causes of traumatic and nontraumatic deaths are presented in Table 1. More than 50% of traumatic deaths were due to collisions with other skiers and objects like trees and rocks, as well as man-made objects such as pylons, fences and a snow groomer. Regarding traumatic deaths, fatal head injuries were the primary cause of death and occurred in 103 victims (54.2%), of whom 63 (61.2%) wore a helmet. A sudden cardiac death (SCD) caused the majority (99.4%) of nontraumatic deaths.

In Table 2, the characteristics of factors associated with traumatic and nontraumatic deaths among fatally injured skiers and snowboarders were shown. Nontraumatic deaths were significantly (*p* = 0.001) higher for males compared to females (93.3% vs. 6.7%). Furthermore, a higher age was significantly associated with nontraumatic death (*p* < 0.001), whereas reported altitude at time of death, used gear and slope difficulty was not significantly different between traumatic and nontraumatic cases (*p* > 0.05).

## 4. Discussion

The aim of the present study was to evaluate incidences and causes of traumatic and nontraumatic deaths among recreational skiers and snowboarders on Austrian ski slopes within a 10-year period from winter seasons 2008/09 to 2017/18. Furthermore, potential differences between traumatic and nontraumatic death cases were investigated.

The mean incidence of fatalities between 2008/09 and 2017/18 was 36.9 ± 7.9 deaths per year. In comparison, studies by Shealy et al. [9] and Ruedl et al. [6] reported 38.5 ± 6.0 and 41.1 ± 6.4 deaths on average per winter season. The overall incidence rate of 0.70 deaths per million skier days is lower than shown in studies by Ruedl et al. [6], Shealy et al. [9] and Bianchi et al. [8], reporting 0.79, 0.75 and 0.70 deaths per million skier days. 

Over the 10-year study period, incidence of traumatic deaths decreased by more than 25%, while the incidence of nontraumatic deaths has reduced by around 40%. 

With regard to Austria, the FMI and the KFAS, concerned with collecting and storing the data, differentiate between traumatic (fall, collision with skier/object, avalanche, etc.) and nontraumatic deaths (sudden cardiac death). Therefore, the present study included all deaths, both traumatic and nontraumatic, while Shealy et al. [9] and Bianchi et al. [8] only documented traumatic deaths. In the underlying study, the incidence of traumatic deaths was 0.36 per million skier days and was much lower compared to incidences reported by studies of Shealy et al. [9] and Bianchi et al. [8]. The abovementioned studies [8,9] analyzed deaths rates in the period from 1991/92 to 2012. Looking at the development of incidences of non-fatal ski injuries at the same time, studies showed a decrease from 5–8 injuries per 1000 skier days before the 1970s [2], to 2–3 injuries per 1000 skier days in the 1980s and early 1990s [3,4] due to advances in equipment and increased helmet use [15]. It might be possible that the incidence of traumatic fatal injuries has benefitted from this decline and from the mentioned equipment-related improvements to a certain extent. Regarding data about Austria, incidences of traumatic (0.36) and nontraumatic (0.34) deaths in the present study were lower compared to results of a previous study by Ruedl et al. [6], who reported an incidence of 0.37 traumatic deaths and 0.42 nontraumatic deaths per million skier days. A potential reason for the lower traumatic death incidences in the present study could be improved skiing ability, as individual skiing skills are related to the injury rate [16,17,18]. In their study, Ekeland et al. [16] reported that beginners had an injury rate three times that of expert skiers. Furthermore, Ekeland et al. [19] showed that skiing ability increased significantly during the selected 12-year from 2000 to 2012.

Looking at the course of overall death incidence, a similar trend was shown among traumatic death incidence. A possible reason could be that traumatic deaths represented the predominant type of death among recreational skiers and snowboarders. On the other hand, a different trend was shown among nontraumatic death incidence compared to the overall incidence. Nevertheless, both, traumatic and nontraumatic death incidence decreased over the 10-year time period, resulting in an incidence of 0.27 deaths per million skier days.

The majority of the available literature showed collisions with solid objects and other skiers to be the most common scenario of a traumatic death on ski slopes [6,7,9]. These results seem to be in line with results of the present study, where more than 50% of traumatic deaths were due to collisions with other skiers and solid and man-made objects. In contrast, a study conducted in Switzerland by Bianchi and Brügger [14] reported a fall during skiing to be the primary cause (47%) of traumatic deaths on ski slopes. Although improvements have been taking place in terms of slope preparation, safety standards on ski slopes and creating awareness of potential risk factors, numbers of skiers are likely to increase and fatal collisions are still happening. According to a study by Ruedl et al. [20], that investigated the causes and factors associated with collisions on ski slopes, preventive recommendations to reduce the risk of a collision include an adaptation of the individual skiing or snowboarding behaviour and the actual speed depending on skill level, weather conditions and the number of other skiers and snowboarders on ski slopes.

It is well known that SCD represents the leading cause for nontraumatic deaths at altitude performing leisure time activities such as recreational skiing [11,12,13]. Confirming these results, the underlying findings represent SCD as the major cause of nontraumatic deaths (99%) on Austrian ski slopes in the last 10 years. Compared to results of a study by Ruedl et al. [6], who stated that SCD was the cause of nontraumatic death in more than 73% of cases, the results of the underlying study are even higher, proofing the existence and importance of this cardiovascular event.

Well in accordance with the study by Ruedl et al. [6], we found that mainly males (87.3%), skiers (95.5%) and persons older than 50 were involved in most deaths. This might be related to the fact that males show, in general, a higher risk-taking behaviour on ski slopes and are skiing faster compared to their female counterparts [1,21], leading to a potentially higher traumatic death rate. Moreover, males show a greater cardiovascular risk of suffering from SCD compared to females [6]. In the present study, the proportion of males was significantly greater in nontraumatic deaths than in traumatic deaths (OR 3.14, 95% CI 1.57–6.27).

Several studies have also reported that males are more prone to fatalities on ski slopes [7,9,12] and traumatic fatality risk is highest between 40–69 years of age [14]. We found that more than 53% of all traumatic deaths were between 40 and 69 years old. With regard to SCD, which causes the majority of nontraumatic deaths, Burtscher and Ponchia [11] already reported an age >34 years as constituting the main risk group. In accordance with this, the results of the present study showed that more than 95% of nontraumatic deaths were older than 34 years. Furthermore, a higher age was significantly associated with nontraumatic death compared to traumatic death (*p* < 0.001). The vast majority of SCD in subjects older than 34 years of age is known mainly to be to due coronary artery disease [22], therefore an increasing frequency of SCDs with age is not surprising [12].

Furthermore, Burtscher [12] showed that SCD seems to be significantly more frequent at altitudes up to 2000 m compared to those over 2000 m. However, further analysis did not detect altitude itself as an important risk factor regarding SCD [12]. Findings of the underlying study seem to be in line with results of Burtscher [12], as mean altitude at death scene was 1552.0 ± 674.1 m above sea level. Moreover, underlying results revealed that altitude at death scene was not significantly different between traumatic and nontraumatic deaths. Burtscher [12] reported that the reason SCDs occurred most frequently at altitudes between 1500 and 2000 m may simply be due to the greater exposure times at that altitude.

Used gear (ski vs. snowboard) and slope difficulty were not significantly different between traumatic and nontraumatic deaths. Interestingly, most traumatic deaths (50%) happened on moderate slopes, whereas the majority of nontraumatic deaths occurred on easy slopes (42%).

The underlying findings suppose deaths on ski slopes are a rare event. Incidences of traumatic and nontraumatic deaths should not be underestimated, as recreational alpine skiing represent one of the most popular winter sports worldwide, with an estimated 400 million skier days annually [23]. However, raising awareness of potential risk factors and common dangers could help reduce fatalities on ski slopes. Based on the currently available evidence, it seems plausible that recreational alpine skiing, especially when performed on a regular basis, may contribute to healthy aging by its association with a healthier lifestyle, including higher levels of physical activity [24].

## 5. Conclusions

In conclusion, incidences of both traumatic and nontraumatic deaths decreased during the 10-year analysis. Nontraumatic deaths were significantly higher among males compared to females. Moreover, increasing age was significantly associated with nontraumatic death, and the majority of traumatic deaths were due to collisions with skiers and objects, whereas SCD caused most nontraumatic deaths. These findings highlight the need of adequate prevention measures to reduce potential risk factors and further decrease the mortality risk on ski slopes.

## Figures and Tables

**Figure 1 ijerph-17-02916-f001:**
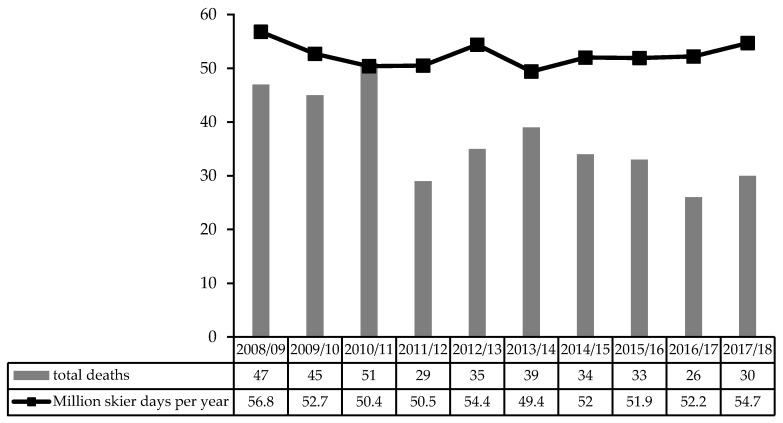
Number of total deaths and million skier days per year from winter season 2008/09 to 2017/18.

**Figure 2 ijerph-17-02916-f002:**
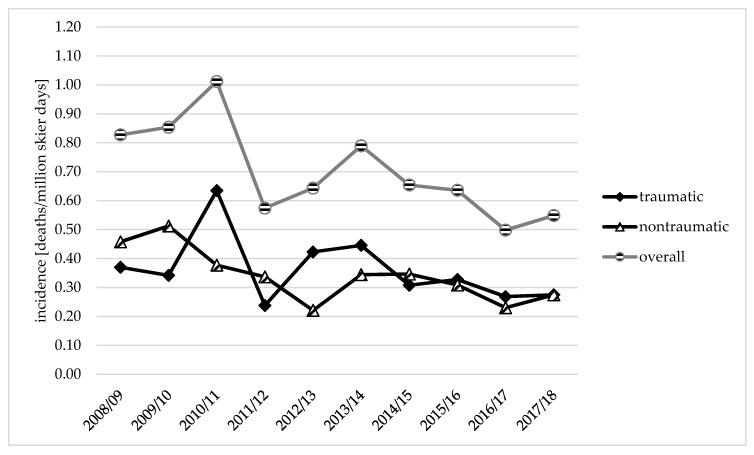
Changes in incidences of traumatic and nontraumatic deaths on Austrian ski slopes in recreational alpine skiers and snowboarders within a 10-year analysis.

**Table 1 ijerph-17-02916-t001:** Causes of traumatic (*n* = 190) and nontraumatic deaths (*n* = 179) on Austrian ski slopes between winter seasons 2008/09 and 2017/18.

Causes of Traumatic Deaths	[*n*, %]	Causes of Nontraumatic Deaths	[*n*, %]
Fall during skiing	89 (46.8)	Sudden cardiac death during skiing	138 (77.0)
Collision with object	76 (40.0)	Sudden cardiac death while standing on the slope	24 (13.4)
Collision with a skier	23 (12.2)	Sudden cardiac death on the lift	15 (8.4)
Avalanche on slope	2 (1.0)	Sudden cardiac death in ski huts	1 (0.6)
		unknown	1 (0.6)

Data are presented as absolute and relative frequencies.

**Table 2 ijerph-17-02916-t002:** Characteristics of factors associated with traumatic and nontraumatic deaths among recreational skiers and snowboarders on Austrian ski slopes between winter seasons 2008/09 and 2017/18.

	Traumatic Deaths (*n* = 190)	Nontraumatic Deaths (*n* = 179)	*p*-Value	Univariate Odds Ratio (95% CI)
**Sex [*n*, %]**				
Male	155 (81.6)	167 (93.3)	0.001	3.14 (1.57–6.27)
Female	35 (18.4)	12 (6.7)		
**Age [years]**	42.8 ± 19.0	59.8 ± 12.7	<0.001	1.07 (1.05–1.09)
**Altitude [m]**	1509.3 ± 640.5	1597.3 ± 707.0	0.106	1.03 (1.01–1.05)
**Gear used [*n*, %]**				
Ski	177 (93.1)	176 (98.3)		
Snowboard	13 (6.9)	3 (1.7)	0.087	0.45 (0.18–1.12)
**Slope difficulty**				
easy (blue)	67 (35.3)	75 (41.9)	0.455	1.18 (0.76–1.84)
moderate (red)	95 (50.0)	72 (40.2)	0.136	0.73 (0.48–1.11)
hard (black)	16 (8.4)	10 (5.6)	0.352	0.68 (0.30–1.54)
unknown	12 (6.3)	22 (12.3)		

Data are presented as mean values ± standard deviation, absolute and relative frequencies and odds ratios (95% CI).

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
