# Peer review of "Incidences of Fatalities on Austrian Ski Slopes: A 10-Year Analysis"

_ijerph, 2020, doi:10.3390/ijerph17082916_

Round 1

Reviewer 1 Report

The present study definitely adds to snowsports Injury Prevention body of knowledge by successfully updating past research on incidences of fatalities on ski slopes, and so it proves most interesting to the readers. It is thorough, well designed, methodologically sound and both results and conclusions clearly presented.

Since I am no native English-speaking I don't feel qualified to soundly comment on the English language and style.

Otherwise I strongly recommend that it be accepted.  

Author Response

Dear reviewer #1,

thanks a lot for your great feedback.

We tried to present additional findings beside the exisiting literature within this topic. Therefore, we are convinced that the underlying paper will receive adequate recognition in the IJERPH.

Best regards,
the authors

Reviewer 2 Report

In table 1: Place the causes in decreasing order of frequencies.

Line 238:  this section??

To compare the traumatic deaths between skiers and snowboarder it would be interesting to know the number of practitioners of each modality or an approximate ratio of them, to know more accurately whether the difference is due to the number of practitioners of the type os activity.

It may also be helpful for the authors to indicate which have been the causes of traumatic deaths in each modality. Although there are few deaths in snowboards, if it were seen that there is a common cause it could prevent further deaths.

Author Response

In table 1: Place the causes in decreasing order of frequencies.

->Thanks for your advice. Now, we placed the causes in decreasing order of frequencies. (line# 127).

Line 238:  this section??

->Thanks a lot. There must have been a typing error. We deleted the phrase “this section” in line# 243.

To compare the traumatic deaths between skiers and snowboarder it would be interesting to know the number of practitioners of each modality or an approximate ratio of them, to know more accurately whether the difference is due to the number of practitioners of the type os activity.

->Thanks a lot for your helpful and interesting comment. As the majority of fatalities occurred in skiers (95.5%) we mainly focused on recreational alpine skiers to provide representative data, which gives a detailed overview of causes of traumatic and nontraumatic deaths. Of course, it would be interesting to know the number of practitioners of each type of activity (skier vs. snowboarder etc.) and to further have a look at potentially diverging proportions of causes of fatalities. Unfortunately, due to the less amount of fatally injured snowboarders (4.5%), underlying data would not be that representative. Furthermore, by focusing on alpine skiers we hoped to make it easier for readers to understand the main findings of this study. Not to forget to mention, we are planning to submit another paper which focuses on the differences in causes of mortalities within winter sport activities (skiing vs. snowboarding).

It may also be helpful for the authors to indicate which have been the causes of traumatic deaths in each modality. Although there are few deaths in snowboards, if it were seen that there is a common cause it could prevent further deaths.

->Thanks a lot for your comment. To make it easy to understand for the readers, we decided to only present the main primary causes of death within traumatic and nontraumatic deaths as there would be too much information. Within line numbers 123-125 we have already reported that “regarding traumatic deaths, fatal head injuries were the primary cause of death and occurred in 103 victims (54.2%) of whom 63 (61.2%) wore a helmet. A sudden cardiac death (SCD) caused the majority (99.4%) of nontraumatic deaths.” These results also contain data of snowboarders. Among snowboarders, also fatal head injuries were the primary cause of death (54.1%) and SCD caused the majority of nontraumatic deaths (99.8%). Therefore, we did not differentiate between snowboarders and skiers and tried to give a clear overview. Furthermore, as mentioned above, we focused on skiers in general to provide representative data, because most fatalities occurred in skiers (95.5%).

Reviewer 3 Report

In general, this is a well written paper. I have only 2 comments:

  • In Abstract, Introduction, Materials and Methods, Results, and Discussion, the authors refer a “10-year period from 2008/09 to 2017/18”. It would be better to change this format because is no clear. For example, I do not understand if “2008/09” means 2008 and 2009 or September 2008.
  • In Table 2, the authors report a p-value of <0.001 for age, but the Confidence Interval (CI) for this variable was omitted. I recommend to include CI for Age and Altitude.

Author Response

In general, this is a well written paper. I have only 2 comments:

In Abstract, Introduction, Materials and Methods, Results, and Discussion, the authors refer a “10-year period from 2008/09 to 2017/18”. It would be better to change this format because is no clear. For example, I do not understand if “2008/09” means 2008 and 2009 or September 2008.

->Thanks a lot for your comment. We always used the figures in conjunction with the term “winter season”. Typically, a winter season starts in the 4th calender quarter of a year and lasts till the second quarter of the upcoming year, depending on the opening times of ski resorts. 2008/09 represents the winter season beginning in the end of 2008 and lasting till the end of the winter season in 2009. This procedure and related specific terminology are usual in the field of epidemiological research in winter sports (please see Ruedl [1,5,6], Ekeland et al [16,19], Posch et al (2019) - https://www.ncbi.nlm.nih.gov/pubmed/30664258; Posch et al (2017) - https://www.ncbi.nlm.nih.gov/pubmed/28445904). We would be very thankful if it is allowed to use these expression. To make the aim of the study clearer, we added the phrase “winter seasons” (line #63,85,150).

 In Table 2, the authors report a p-value of <0.001 for age, but the Confidence Interval (CI) for this variable was omitted. I recommend to include CI for Age and Altitude.

->Thanks a lot for your helpful comment. We added CI for the variables age and altitude (see line#143).